# Research on Dynamic Risk Assessment and Active Defense Strategy of Active Distribution Network under Ice Weather

**Xin-Rui Liu [1],\*, Hao Wang [1], Qiu-Ye Sun [1] and Peng Jin [2]**

[1] Department of Electrical Engineering, College of Information Science and Engineering, Northeastern University, Shenyang 110819, China; 1870498@stu.neu.edu.cn (H.W.); sunqiuye@ise.neu.edu.cn (Q.-Y.S.)

[2] State Grid Liaoning Electric Power Supply Co. LTD., Shenyang 110003, China; jp@ln.sgcc.com.cn

\* Correspondence: liuxinrui@ise.neu.edu.cn; Tel.: +86-24-8368-3907

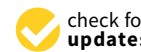

**Featured Application: This paper has potential applications in the monitoring of ice disaster distribution network operation status and prevention of power accidents based on defense strategies.**

**Abstract:** With the continuous development of the active distribution network (ADN), the problem of security and stability has become increasingly prominent. From the perspective of improving the defense capability of ADN, a new a multi-angle dynamic risk assessment index system based on the comprehensive vulnerability rate model is proposed in this paper. Risk threshold is used to monitor the status of the distribution network, which determine whether ADN needs to enter the active defense period. The minimum amount of load shedding outside the fault isolation region is regarded as the objective function, considering other constraints such as limited resources, the coordinated active defense strategy (CADS) is formed in this paper. Finally, the accuracy of the comprehensive vulnerability rate and the risk assessment value are verified by example analysis, and the superiority of the CADS is verified by comparing different defense strategies.

**Keywords:** active distribution network; comprehensive vulnerability rate; multisource situational awareness; risk assessment; active defense; load transfer; data fusion; islands

## 1. Introduction

As one of the most extreme weather, the ice disaster will pose a great threat to the safe and stable operation of the distribution network [1,2]. There have been a lot of work about transmission line and distribution line in icing weather. Ice accretion can cause faults such as line breaking, tower falling, line dancing, insulator flashover [3,4]. In [5], a simple ice-heating equation for ice-covered surface was established to reveal the relationship between ice coating and thermodynamics of transmission lines. All of the above works have studied the influence of the external environment on the line failure rate. In addition, the instability of the internal operating conditions of the distribution network caused by the extreme disaster environment is also an important cause of failure [6].

The distribution network safety assessment methods mainly include reliability assessment and risk assessment [7–10]. The reliability assessment is to verify whether the power system facilities or grid structure meet the specified reliability criteria, which is mostly used for grid planning. The evaluation method based on dynamic risk theory can evaluate the operation status of the distribution network in real time and analyze the vulnerable links, so it has superiority compared with reliability analysis. In the past, most of the distribution network risk assessment models were based on the product of the probability of failure and the severity of the consequences [11,12] instead of more

reasonable evaluation models, and there is not formed a perfect risk assessment evaluation index system of the ADN in the ice weather yet. In [13], a model based on ice faults in clusters was proposed which introduced some vulnerability indexes to analyze the risk of future ice storms.

Furthermore, related research on distribution network fault recovery strategies is also common. However, to date, most of them focus on the method of repair and the post-disaster recovery process after the accident [14–16], there is a lack of active defense strategy based on fault prediction. Some relevant ideas and technical analysis of security defense were proposed in [17–20]. One method is the strategy based on DC ice-melting technology in icing weather, which is difficult to guarantee efficiency because it is only applicable to transmission lines or fault areas with ice melting devices but no effective defense and protection for loads outside the fault region. The other method is use matching plans as a method based on historical data and typical situations, this model cannot meet the requirements of real-time online decision-making, the information utilization is very low, and there is a lack of coordinated ability for the local and global distribution network by using the defense strategy, which cannot guarantee efficient use of resources and the maximum power supply.

Therefore, in order to improve the adaptability of active distribution network in ice weather, the main contributions are introduced from three aspects in this paper:

1.  The multisource situational awareness method is used to analyze the internal operation situation of the distribution network in this paper, define the network topology situation, the power supply situation, the defense resource, the power flow situation, and cascading fault factor, thus the comprehensive vulnerability rate model is established.
2.  An ice hazard dynamic risk assessment index system concluding the ice-covered dynamic intrusion index, ice disaster-resistance capability index and demand response potential index is proposed in this paper so that it can monitor the current operating status of the distribution network in real time. In order to evaluate the risk status of different distribution lines at a certain moment and determine whether a certain line needs to take active defense plan at the current moment, the concept of risk threshold is proposed in this paper.
3.  Making full use of the transfer capacity of the tie-lines and the flexible controllability of distributed energy such distributed generations (DGs) and energy storage system (ESS), an evolutionary process of coordinated active defense strategy (CADS) based on fault prediction which involves load transfer with DGs connected in active distribution network (ADN), island operation, and load shedding is established in this paper. The minimum amount of load shedding is considered as an optimization goal while considering various constraints, the active defense scheme is more realistic and the practicality of the active defense strategy is improved.

The rest of this paper is organized as follows. Section 2 is the vulnerability analysis of distribution network based on fault pre-judgment. Section 3 is a multi-angle dynamic risk assessment model. Section 4 conducts research on active defense strategies (CADS). Analysis and simulations were performed in the paper in Section 5. Section 6 summarizes the paper.

## 2. The Comprehensive Vulnerability Rate Model

### 2.1. Ice Accretion Model

The ice weight model can be obtained in [21], as shown in Equation (1). The modified ice accretion model is obtained by considering the collision coefficient $\alpha_1^j$ and the freezing coefficient $\beta_j$, as shown in Equation (3):

$$M_j = \int_0^l \left[ \frac{4}{\pi} R(t) \left( 1 + \frac{C_a V_a}{V_j(t) R} \right)^{-1} V_j(t) W_j(t) C_1 sin\theta_1 sin\theta_2 \right] dt \tag{1}$$

$$W_j(t) = 0.067 \rho_0 S_j(t)^{0.846} \tag{2}$$

$$\Delta R(t) = \frac{0.035}{1 + \frac{21.65}{V_j(t)D_j(t)}} \alpha_1^j \beta_j \sqrt{(\rho_0 S_j(t))^2 + (3.6 V_j(t) W_j(t))^2} \tag{3}$$

where $S_j(t)$ is the rainfall in one hour, the unit is mm/h. $W_j(t)$ is the average water content per cubic meter in the air. $V_j(t)$ is the average wind speed. The diameter of the supercooled water droplet is $D_j(t)$. The ice constant is $C_a$. $V_a$ is viscosity of air movement. $C_1$ is the collection coefficient. The angle between the raindrop and distribution lines is set as $\theta_1$, and the angle between wind and distribution lines as $\theta_2$. $R$ is the radius of the distribution line in meters. $\rho_0$ is the density of freezing rain or supercooled water, the unit is $kg/m^3$.

## 2.2. Ice Accretion Failure Rate Model

Calculating different types of failure rates is the primary step in risk assessment. Fault events based on icing weather are mainly mechanical faults and electrical faults, which consist of five events: no event $(Y_1)$, distribution line disconnection $(Y_2)$, inverted tower $(Y_3)$, distribution line dancing $(Y_4)$, and insulator flashover $(Y_5)$. The fault rate in ice weather is conditional probability. The probability of different events is calculated according to parameters such as ice accretion condition, wind load, critical voltage of insulator, and actual running voltage combined with the difference between the parameters of the lines and the micro-topography and micro-meteorology [22]. The 66 kV overhead distribution line is taken as the research object in this paper. The probability of the icing events can be formulated as:

$$P_k^t(Y_i) = 1 - \prod_{j=1}^{z} \left(1 - P_{kj}^t(Y_i)\right) \quad i = 2, 3, 4, 5 \tag{4}$$

Set $Z$ is the span of the distribution line, the probability of no fault event, line disconnection event, inverted tower event, line dancing event, and ice flashover event of the distribution line $k$ is $P_k^t(Y_1)$, $P_k^t(Y_2)$, $P_k^t(Y_3)$, $P_k^t(Y_4)$, $P_k^t(Y_5)$, then $P_k^t(Y_1)$ can be calculated by the following formula:

$$P_k^t(Y_1) = 1 - \sum_{a=2}^{5} P_k^t(Y_i) \tag{5}$$

Finally, the definition of "faults directly related to icing events" is event $X$, according to the full probability formula, the ice failure rate of the distribution line as follows:

$$P_k^t(X) = \sum_{a=1}^{5} P(X|Y_i) \cdot P_k^t(Y_i) \tag{6}$$

These events are not only related to the ice accretion condition of the line, but also involve complex analysis processes such as mechanics, meteorology and aerodynamics. In the calculation, a large amount of measurement data, grid component design parameters, state parameters, etc. are required. At the same time, it is limited by the distribution network related monitoring system. Therefore, a simplified calculation based on the existing reference and actual conditions is used in this paper. See Appendix A for details.

## 2.3. Multisource Situational Awareness Analysis

The failure rate of distribution line under ice disaster is not only a model which changes with ice accretion condition, but also closely related to various status information of the distribution network such as network topology, power supply equipment, and network defense resources. Therefore, collecting various data about the internal operating conditions and performing fusion analysis in the ice weather can more accurately analyze and predict the vulnerable links of the distribution network, thus significantly improve the early warning capability of the risk.

### 2.3.1. Network Topology Situation

This indicator reflects the importance of the distribution line in the distribution network topology. The power supply in the area is set as $c_1$, the number of loads on distribution line $k$ is set as $c_2$, the total loads in the area is set as $c_3$, $Z_k$ is the impedance of the distribution line $k$, and $N_{st}$ is the optimal numbers of paths from power $S$ to load $T$, $N_{st}(k)$ is the number of times the optimal path between the power supply $S$ and the load $T$ passes through the distribution line $k$. $\eta_{k,1}^t$ is the network topology situation, which is used to describe the contribution of the distribution line to the power flow in the whole network.

$$\eta_{k,1}^t = Z_k \cdot \frac{\sum_{c_1}\left[\sum_{c_2} N_{st}(k)\right]}{\sum_{c_1}\left[\sum_{c_3} N_{st}\right]} \tag{7}$$

### 2.3.2. Power Supply Situation

This indicator reflects the impact of power supply equipment on distribution line operation. Power distribution equipment includes conventional power supply equipment and various types of distributed energy equipment. The external environment has different effects on their operation.

(1)  Conventional power supply

As the period increases, the failure rate and the power supply risk that most of the conventional power components such as transformers are also increased. The failure rate of a power component is described by Weibull function, as shown in Figure 1. If the equipment maintenance interval is $\Delta t$ in the loss period, and the failure rate after maintenance is reduced from $\lambda_i$ to $\mu_i$ times the basic failure rate, the equipment failure rate $\lambda_z$ is:

$$\lambda_z = \mu_i \lambda_L + \frac{\beta_{2i}}{\tau_{2i}^{\beta_{2i}}}(t - t_i)^{\beta_{2i}-1} \tag{8}$$

where $\tau_{2i}$ is the scale parameter. $\beta_{2i}$ is the shape parameter, which indicate the influence of the repair on the basic failure rate and the aging of the component.

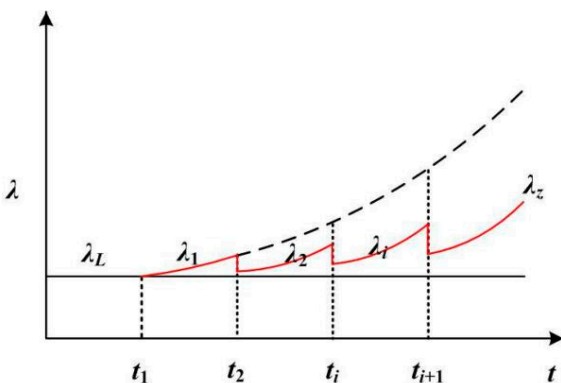

**Figure 1.** Equivalent curve of actual aging failure rate.

Thus, the power supply index (PSI) of the conventional power supply system is:

$$\lambda_k = \frac{\sum_{m=1,\cdots,a_{1,k}} |\varepsilon_{m.g}| \lambda_{m.z}}{\sum_{m=1,\cdots,b_1} \lambda_{m.z} + \sum_{m=1,\cdots,b_2} P_{m.dg} + \sum_{m=1,\cdots,b_3} P_{m.ess}} \tag{9}$$

(2)  Distributed generations (DGs)

Icing weather has a great impact on the output of distributed power, the wind power DG and the photovoltaic power output model is as follows:

$$P_f(v) = \begin{cases} 0 & v \leq v_q \\ \frac{P_r}{v_r - v_q} v - \frac{P_r}{v_r - v_q} v_q & v_q < v < v_r \\ P_r & v_r < v < v_r \\ 0 & v \geq v_t \end{cases} \tag{10}$$

$$P_{ph} = A \cdot S \cdot \phi \cdot \eta_0 [1 - 0.005 \cdot (T - 25)] \tag{11}$$

where $P_r$ is power rated of the wind turbine. $v_r$ is the rated wind speed. $v_q$ is the cut-in wind speed. $v_t$ is the cut-out wind speed. $A$ is the number of photovoltaic panels. $S$ is the area of a single photovoltaic panel. $\phi$ is the solar radiation intensity. $\eta_0$ is the photoelectric conversion efficiency under standard test conditions. $T$ is the ambient temperature, the unit is °C.

Thus the power supply index (PSI) of DGs is:

$$\delta_k = \frac{\sum_{m=1,\cdots,a_{2,k}} |\varepsilon_{m.dg}| P_{m.dg}}{\sum_{m=1,\cdots,b_1} \lambda_{m.z} + \sum_{m=1,\cdots,b_2} P_{m.dg} + \sum_{m=1,\cdots,b_3} P_{m.ess}} \tag{12}$$

(3)　Energy storage system (ESS)

It is assumed that the ESS does not participate in power supply when it is connected to ADN, so only the power supply situation of the ESS in the island is considered in this paper. When the faults occur in the ADN, the area that is powered by the backup power supply (BPS) is called an island. With the development of ESS technology, it plays a more important role in the ADN. Therefore, this paper argues that the BPS consists of the ESS and DGs. Among them, the BPS in the island may include multiple ESSs and DGs. The model of ESS is as follows and constraints of ESS divided into state of charge (SOC) constraints, maximum charge, and discharge constraints.

$$\begin{cases} E_{j,bat}^{t+1} = E_{j,bat}^{t} + P_{j,ch}^{t} \eta^{ch} T - \frac{P_{j,dis}^{t} T}{\eta^{dis}} \\ E_{j,bat,max} \times 20\% \leq E_{j,bat}^{t+1} \leq E_{j,bat,max} \times 90\% \\ E_{j,bat,max} \times 20\% \leq E_{j,bat}^{t} \leq E_{j,bat,max} \times 90\% \quad t \in [2, N_T] \\ 0 \leq P_{j,dis}^{t} \leq D_{j,dis}^{t} P_{j,dis,max}^{t} \\ 0 \leq P_{j,ch}^{t} \leq D_{j,ch}^{t} P_{j,ch,max}^{t} \\ D_{j,dis}^{t} + D_{j,ch}^{t} \leq 1 \forall t \in [1, N_T], \forall j \in [1, N_{bus}] \end{cases} \tag{13}$$

where, $E_{j,bat}^{t+1}$ is the remaining capacity of the node $j$ energy storage device at the end of the period of $t+1$. $P_{j,ch}^{t}$ and $P_{j,dis}^{t}$ represent the charge and discharge power of the ESS in the period $t$, respectively. $\eta^{ch}$ and $\eta^{dis}$ are charge and discharge efficiency. $E_{j,bat,max}$ indicates the maximum capacity of the ESS. $P_{j,ch,max}^{t}$ and $P_{j,dis,max}^{t}$ are the maximum charge and discharge powers of the node $j$ in the ESS. $D$ is a 0–1 variable and $D_{j,ch}^{t}$ and $D_{j,dis}^{t}$ indicate the state of charge and discharge at time $t$. $T$ is the duration of a risk assessment. $N_T$ is the total number of risk assessment time intervals. $N_{bus}$ is the number of nodes in the ADN.

Then the power supply index (PSI) of the distributed ESS for distribution line $k$ is:

$$\gamma_k = \frac{\sum_{m=1,\cdots,a_{3,k}} |\varepsilon_{m.ess}| P_{m.ess}}{\sum_{m=1,\cdots,b_1} \lambda_{m.z} + \sum_{m=1,\cdots,b_2} P_{m.dg} + \sum_{m=1,\cdots,b_3} P_{m.ess}} \tag{14}$$

In order to distinguish the influence of the different power supply equipment status on the operation of the distribution lines, the power contribution tracking coefficient of the transformer, wind

power photovoltaic equipment and energy storage device to the distribution lines can be obtained by the power flow tracking, which are $\varepsilon_{m.g}$, $\varepsilon_{m.dg}$, $\varepsilon_{m.ess}$ respectively. The number of transformers, wind power photovoltaic equipment and distributed energy storage devices on the distribution line are $a_{1,k}$, $a_{2,k}$, $a_{3,k}$ respectively. The number of transformers, wind power photovoltaic equipment, and distributed energy storage devices in the distribution network area are $b_1$, $b_2$, $b_3$ respectively. Then the power supply equipment risk situation index of the line can be expressed as:

$$\eta_{k,2}^t = 1 - (\tau_1 \cdot \lambda_k + \tau_2 \cdot \delta_k + \tau_3 \cdot \gamma_k) \tag{15}$$

where $\tau_1$, $\tau_2$, and $\tau_3$ are the weight.

### 2.3.3. Defense Resources

This indicator reflects the defense capabilities of the defense resources of the distribution network, including staffing of emergency repair, emergency equipment resources (including emergency vehicles, lifting equipment, safety equipment, etc.). Suppose a defense resource owned by a distribution line is $u$, $U^t$ is the real-time available defense resources in the distribution network. The user's power protection level on the distribution line is w, the value is set to 4 levels from high to low according to the importance level, which is expressed as $1 \geq w_1 > w_2 > w_3 > w_4 \geq 0$. The defense resources situation index of the distribution line is:

$$\eta_{k,3}^t = \frac{w}{1 + 0.25 u_k^t} \tag{16}$$

$$\sum_{i=1}^{m} u_k^t = U^t \tag{17}$$

### 2.3.4. Power Flow Situation

Distribution line faults in the icing weather may cause redistribution of the power flow which cause some components to exit. In the distribution system, the line failure rate has the following relationship with the power flow [23]:

$$\eta_{k,4}^t = \begin{cases} P_0^t & 0 \leq l \leq L_{nor} \\ P_0^t + \frac{(1 - P_0^t)(l - L_{nor})}{L_{max} - L_{nor}} & L_{nor} \leq l \leq L_{max} \\ 1 & l \geq L_{max} \end{cases} \tag{18}$$

The model shows that if the active power does not reach the rated capacity, the line runs normally, and the possibility of line fault is stable at a small probability value $P_0^t$. If the active power is between the rated capacity and the maximum capacity, the probability of failure shows a linear growth trend, $L_{nor}$ is the rating power of the distribution line, $L_{max}$ is the power limit of the distribution line. If the active power exceeds the limit of the line transmission, the failure rate is 1.

### 2.3.5. Cascading Faults Factor

The cascading faults of the distribution line in the ice weather is related to the influence of the power flow. The power flow disturbance will make some distribution lines in the self-organized critical state under certain conditions [24], according to the theory of complex systems. It is easy to cause cascading faults when the power system is in a self-organized critical state. The criteria for judging a critical line is as follows:

$$H = \frac{|L_i| / L_{i,max}}{\sum_I |P_{li}| / \sum_I P_{li,max}} \geq \alpha \tag{19}$$

$$\sum_I |P_{li}| \epsilon (0.56 - 1.00) \sum_I P_{li,max}$$

where $H$ is the critical coefficient, $L_{i,max}$ is the rated power, $L_i$ is the actual power, $P_{li}$ is the active power flow of the branch, $P_{li,max}$ is the rated active transmission capacity of the branch, $I$ is composed of all lines and transformers set, $\alpha$ is the critical reference coefficient which the empirical value is generally 1.05.

Let the total number of lines in the area be $N$, and the chain failure propagation degree of the long-distance connection line $k$ is defined as:

$$g_k = \frac{N_{count} \sum_{y=1,\cdots,s} f(k,y)}{N_{count} \sum_{i=1,\cdots,N} k} \tag{20}$$

where $N_{count}$ is the counting function. $N_{count} \sum_{y=1,\cdots,s} f(k,y)$ is the correlation between distribution line $k$ and distribution line $y$. If $k$ fails, $y$ also fails, meaning they are related [25], $f(k,y) = 1$, otherwise $f(k,y) = 0$. $N_{count} \sum_{i=1,\cdots,N} k$ is the number of all distribution lines associated with distribution line $k$.

Define the average failure rate of the ice disaster is $R_{ax} = n_2/n_{a2}$, then cascading faults situation index of the distribution line $k$ that considering the power flow disturbance is as follows:

$$\eta_{k,5}^t = R_{ax} \cdot g_k \tag{21}$$

Any extreme situation index value may cause distribution fault. In order to meet the actual situation, standard deviation is proposed in this paper to jointly describe the impact of the internal operating situation on the failure rate of the distribution network. Thus, use Equation (22) to determine the number of useful situation indicators $N$ from $\eta_{k,i}^t \in \{\eta_{k,1}^t, \eta_{k,2}^t, \eta_{k,3}^t, \eta_{k,4}^t, \eta_{k,5}^t\}$. The initial value of $N$ is 5. $\sigma_0$ is critical value of the standard deviation. If $\sigma > \sigma_0$, the smallest indicator is ignored and the value of $N$ decreases by 1. Repeat the verification by Equation (22) until $\sigma \leq \sigma_0$ and determine the value of $N$. Finally, the internal operation situation factors $\eta_{k,sit}^t$ can be calculated by using Equation (23).

$$\sigma = \sqrt{\frac{1}{N} \sum_{i=1}^{N} \left(\eta_{k,i}^t - \overline{\eta_k^t}\right)^2} \tag{22}$$

$$\eta_{k,sit}^t = exp\left[\sum_{i=1}^{N} \theta_i \ln \eta_{k,i}^t\right] \tag{23}$$

where $\theta_i$ indicates the index weight, $\sum \theta_i = 1$.

*2.4. Comprehensive Vulnerability Rate*

It is assumed that the distribution line failure rate directly affected by ice accretion and the internal operation situation factors are two independent parts, thus the comprehensive vulnerability rate model under the ADN in the ice weather is:

$$F(k,t) = 1 - \left(1 - P_k^t(X)\right)\left(1 - \eta_{k,sit}^t\right) \tag{24}$$

## 3. Multi-Angle Dynamic Risk Assessment

Risk assessment is an important basis for the active defense of the distribution network. According to the severity of the risk assessment, the dispatching department can determine the corresponding emergency measures. This paper attempts to construct a multi-angle risk assessment index system through three aspects: ice-covered dynamic intrusion index, ice disaster-resistance capability index and demand response potential index [26,27]. Among them, ice-covered dynamic intrusion index reflects

the ice-covered degree and the development trend of the ice disaster of the distribution network. Ice disaster-resistance capability index reflects the anti-interference ability of the distribution line under the current environmental conditions. Demand response potential index can be understood as the defense capability of the distribution network, including the defend resources and the ability to melt ice and deicing. The multi-angle risk assessment index system of the ADN in the ice weather is shown in Figure 2.

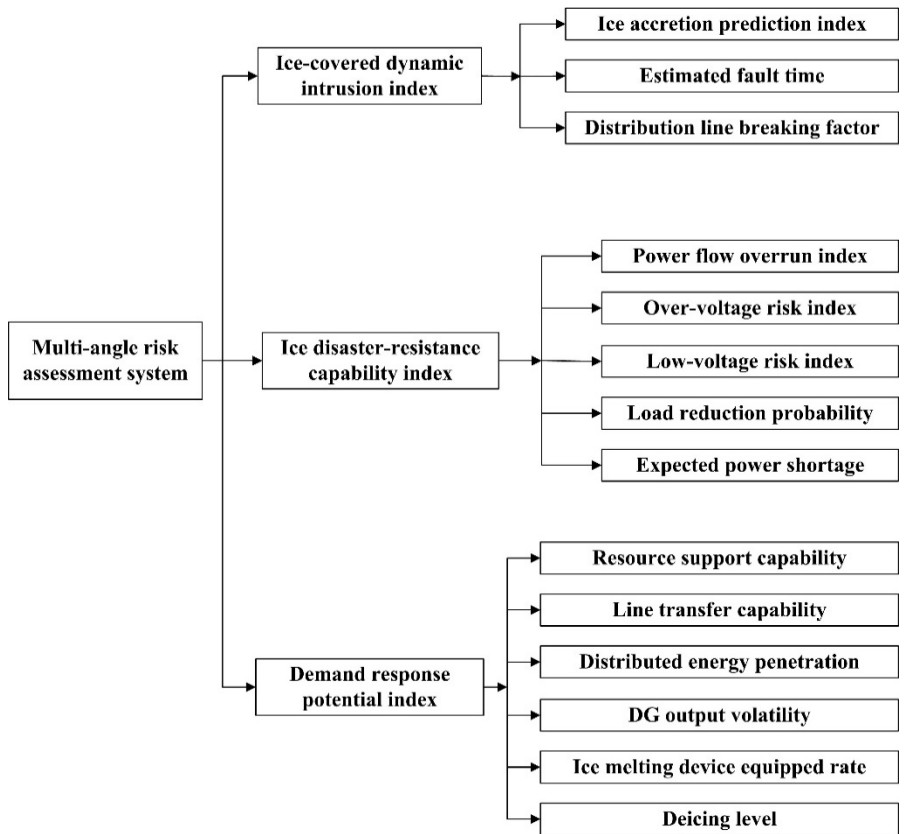

**Figure 2.** Multi-angle risk assessment system.

*3.1. Ice-Covered Dynamic Intrusion Index*

It includes three aspects: ice accretion prediction index, estimated fault time, and distribution line breaking factor, which attempt to describe the state of distribution network from ice weather. With the changes of micro-meteorological parameters such as precipitation, wind speed, temperature, and wind direction, each predicted value will gradually become the old data. Hence, there is an error between the actual ice accretion value $R_r(t_{i-1})$ and the ice accretion prediction value $R_r(t_{i-1})$ at the previous moment. The ice accretion prediction model $R(t_i)$ in the next moment can be corrected by using the ice accretion prediction value matching index $\delta(t_{i-1})$ at the previous moment. Then the predicted ice accretion at time $t$ can be calculated according to the following formula:

$$\delta(t_{i-1}) = \frac{R(t_{i-1})}{R_r(t_{i-1})} \times 100\% \tag{25}$$

$$R(t_i) = R_r(t_{i-1}) + \int_{t_{i-1}}^{t_i} \frac{\Delta R(t)}{\delta(t_{i-1})} dt \tag{26}$$

Additionally, if no defensive strategy can be taken in icing weather, it makes it possible to cause fault after some time. Hence estimated fault time is as follows:

$$T_{fault} = \frac{1}{\left[\left|0.133\frac{R}{dl} - 0.011\right| \times v_0\right]} \tag{27}$$

where $R$ is the predicted value of ice thickness, $dl$ is the design value of distribution line ice thickness, and $v_0$ is the influence factor of line length.

Assume the predicted ice weight value is $M(t_1)$, $M_i(t_2) \dots M_i(t_n)$ in the whole process, set $F_1$ and $F_2$ are the wind load caused by the horizontal and vertical winds. $G$ is the self-heavy load of the distribution line, thus the predicted value of total load $H$ can be calculated by the following formula:

$$H = \left[(M + F_2 + G)^2 + F_1^2\right]^{\frac{1}{2}} \tag{28}$$

Due to the maximum stress that can be withstood by each distribution line is not the same, set the break factor $\phi$:

$$\phi = \frac{H}{N_{max}} \tag{29}$$

### 3.2. Ice Disaster-Resistance Capability Index

According to network topology situation from Section 2.3 combined with the comprehensive vulnerability rate from Section 2.4, the ice disaster-resistance capability can be evaluated by power flow overrun index $r_a$, over-voltage risk index $r_b$, low-voltage risk index $r_c$, load reduction probability $\xi(k,t)$, and expected power shortage $C(k,t)$:

$$\begin{cases} r_a = \eta_{k,4}^t \cdot \left|\frac{L_{t,k} - L_{t,o}}{L_{t,o}} \times \eta_{k,1}^t\right| \\ r_b = F(k,t)\sum_j(\frac{U_{t,j} - U_{t,o}}{U_{t,o}} \times O_j) \\ r_c = F(k,t)\sum_j(\frac{U_{t,u} - U_{t,j}}{U_{t,u}} \times O_j) \end{cases} \tag{30}$$

where $L_{t,i}$ is the power flow of the distribution line $k$ at time $t$. $L_{t,o}$ is the power flow limit of the distribution line $k$. $U_{t,j}$ is voltage of the node $j$ at time $t$. $U_{t,o}$ and $U_{t,u}$ are the upper and lower limits of voltage at time $t$. $O_j$ is the importance index of the node $j$.

$$\xi(k,t) = \sum_{c=1}^{p} \frac{t_i}{T}F(k,t_i) \cdot \eta_{k,1}^t \tag{31}$$

$$C(k,t) = \sum_{c=1}^{p} t_i \cdot exp[F(k,t_i)] \cdot \eta_{k,1}^t \tag{32}$$

where $T$ is the duration of the entire ice weather, and $t_i$ is a period in which under different power flow and demand of loads.

### 3.3. Demand Response Potential Index

Demand response potential index includes resource support capability $e_1$, tie-line transfer capability $e_2$, distributed energy penetration $e_3$, DGs output volatility $e_4$, ice melting equipment rate $e_5$, and deicing rate $e_6$, as provided in Equation (33).

$$\begin{cases} e_1 = \frac{E_i}{\sum_c E_j} \times 100\% \\ e_2 = \frac{\sum I_i}{\sum_c I_j} \times 100\% \\ e_3 = \frac{\sum P_{res}(t)}{P_L(t)} \times 100\% \\ e_4 = \frac{\sqrt{\frac{\sum_{i=1}^n \{P_{res}[(i+1)\times\Delta T] - P_{res}(i\times\Delta T)\}^2}{n}}}{P_{res}^*} \times 100\% \\ e_5 = \frac{q_i}{\sum_c q_j} \times 100\% \\ e_6 = \frac{\sum w_{i,s}\times\mu_s}{\sum_c q_j} \times 100\% \end{cases} \tag{33}$$

In the formulas: $E_i$ is the number of resources allocated to the distribution line. $\sum_C E_j$ is the total amount of resources in the region. $\sum I_i$ is the number of the tie-lines that can be transferred for the distribution line. $P_{res}(t)$ is the actual output of the distributed energy power supply connected to the distribution network. $P_L(t)$ is the actual loads value of the distribution network. $\Delta T$ is the reference time interval. $i$ is the number of reference time intervals corresponding to this moment. $n$ is the total number of all-day intervals. $P_{res}(i \times \Delta T)$ is the actual output of the distributed energy at previous moment. $P_{res}[(i + 1) \times \Delta T]$ is the actual output of the distributed energy in next moment. $P_{res}^*$ is the distributed energy rated power in the network. $q_i$ is the number of ice melting devices on the distribution line. $w_{i,s}$ is the deicing device $S$ of the distribution line. $\mu_s$ is the action coefficient of the deicing device $S$.

### 3.4. Dynamic Risk Assessment Model

In order to measure the whole operating state of the system comprehensively and improve the evaluation sensitivity of the network security situation, it is necessary to integrate the various sub-indicators to establish a dynamic risk assessment model. Analytic hierarchy process (AHP) is a comprehensive decision-making method for solving multi-objective problems. It belongs to subjective evaluation method. The entropy weight method (EWM) is an objective evaluation method based on information entropy principle, which belongs to objective evaluation method. In this paper, the analytic hierarchy process-entropy weight method (AHP-EWM) [28] is proposed based on the advantages of these two methods, which makes up for the shortcomings of the single method and improves the accuracy of the risk evaluation index weights. In order to make the weight deviation of these two methods small, a least squares optimization combination weight model is proposed in this paper. Assume that the weights of the indicators obtained by the analytic hierarchy process and the entropy weight method are $\omega_j$ and $\varphi_j$, the model is established as follows:

$$\begin{cases} min\, F(\mu) = \sum_{i=1}^m \sum_{j=1}^n \left\{ \left[(\omega_j - \mu_j)s_{ij}\right]^2 + \left[(\varphi_j - \mu_j)s_{ij}\right]^2 \right\} \\ \sum_{j=1}^n \mu_j = 1,\, \mu_j \geq 0(j = 1,2,\ldots,n) \end{cases} \tag{34}$$

The comprehensive weight value $\mu_j$ of each index can be obtained by using the Lagrange multiplier method. $s_{kj}^t$ is the $j$-th index value of the distribution line $k$ at time $t$. Then the risk assessment model based on the comprehensive weights is as follows:

$$L^t = \begin{bmatrix} \mu_1, & \mu_2, & \ldots, & \mu_n \end{bmatrix} \times \begin{bmatrix} s_{11}^t & s_{12}^t & \cdots & s_{1n}^t \\ s_{21}^t & s_{22}^t & \cdots & s_{2n}^t \\ \cdots & \cdots & \cdots & \cdots \\ s_{m1}^t & s_{m2}^t & \cdots & s_{mn}^t \end{bmatrix}^T \tag{35}$$

### 3.5. Risk Threshold

As an important evaluation standard for distribution network risk assessment, the risk threshold plays an important role in the distribution risk judgment of abnormal. The operating status of the ADN can be divided into three categories, namely: normal operation status, risk warning status, and fault status. The critical value between the normal operating status and the risk warning status is called the risk threshold. If the current operational risk value is less than the threshold, the distribution network is in normal operation. If the risk value is greater than the threshold which makes the frequency of operation of the distribution network is abnormal, the voltage exceeds the limit and the power is insufficient, it indicates that the line is at high-risk state and the defense strategy must be taken to prevent the risk expanding and ensure the continuous power supply of the important load to the maximum extent. The risk threshold for the distribution line is as follows:

$$V_k = \left(1 - \frac{\eta_{k,1}^t}{\sum_s \eta_{i,1}^t}\right) \times V_0 \tag{36}$$

where $V_k$ is the risk threshold of the distribution line. $V_0$ is the standard threshold of the distribution line, which generally between 0.55–0.6. If $R_k > V_k$, it means that risk value of the distribution line $k$ is in the risk warning status.

## 4. CADS

The research object of active defense is the high-risk power loss areas outside the fault areas. The purpose of active defense is to protect important loads as much as possible and prevent the expansion of power loss areas. Based on the risk assessment, the minimum load removal is set as the objective function in this paper, combined with the constraints of limited resources, the CADS which contains islands, load transfer with DGs connected in ADN and load shedding is established.

### 4.1. Load Transfer with DGs Connected in ADN

In this paper, the load transfer is a power supply protection strategy based on fault pre-judgment. It is effective to avoid the accident expansion and reduce the loss of loads by changing the states of switches when the distribution line is in an abnormal situation. At the same time, the contribution of distributed energy to the active defense mechanism should be considered. Hence, the DGs can be used when the loads of the to-be-transferred is greater than the capacity of the tie-line because the cost of switch operation of this scheme is usually less than island mode [29,30]. However, it is must be satisfied that the power flow direction of DG and the tie-line are same, otherwise the island mode is considered. Therefore, the goal of the load transfer with DGs connected in ADN is:

$$\begin{cases} maxP_{Trans} = \sum_s P_{i,tie} + P_{DE} \\ minN = min\left[\sum_{i=1}^a (1 - S_i) + \sum_{j=1}^b T_j\right] \end{cases} \tag{37}$$

where, $P_{Trans}$ is the amount of loads which can be transferred by using the strategy. $P_{i,tie}$ is the capacities of all available tie lines of the distribution line. $P_{DE}$ is the distributed energy which can supply for the distribution line. $N$ is the number of switching operations, $S_i = 1$ is the section switch action, $S_i = 0$ is no action. $T_j = 1$ is that the interconnection switch operates during the process, and $T_j = 0$ is that no action. Subject to:

$$\begin{cases} V_{i,lowm} \leq V_i(t) \leq V_{i,high}, \forall i, t \\ -I_{b,min} \leq I_b(t) \leq I_{b,max}, \forall b, t \\ g_i \in G \end{cases} \tag{38}$$

Among them, constraint (38) limits the voltage, branch power flow, and distribution network radial.

*4.2. Island Operation*

When the distribution line is equipped with BPS, it can take the way of isolated islands to ensure the supply of some important loads [31]. The BPS in the island is divided into two types: ESS and DGs. In order to simplify the analysis process, the paper uses one ESS and multiple DGs as examples to analyze the model of the island. Due to the DGs output has the characteristics of randomness and fluctuation, it must rely on ESS to adjust the power balance of the entire island. Therefore, this paper proposes the island model from three steps.

1. The first step determines the feasibility of the island model according to the defense judgment conditions. Due to the field of research in this paper is the defense plan before occur faults, it must be ensured that the island in the defense process is feasible and meaningful. In other words, the island mode fails if the following conditions (1) or (2) are true in the CADS (coordinated active defense strategy).

(1)    A high-risk line is directly connected to the root node of the island and the direction of the power flow is from the root node to the high-risk line.
(2)    During the entire active defense process, DGs and ESS are unavailable. That is, $P_m$ is less than the minimum power support capability.

If condition (1) holds, once the high-risk line fails, the nodes downstream of the line cannot be protected by the island and may even bring greater load loss. If condition (2) holds, the BPS cannot provide enough power to the load, it is obviously impossible to use island mode. These conditions will be an important part of the partition of the island.

2. If neither 1 nor 2 is true, the second step is performed which uses the Tree Knapsack Problem (TKP) to determine the initial island range, the goal and constraint of island range is:

$$
\begin{cases}
max\ P_{island} = \sum_{i=1}^{T_n} \sum_{j=1}^{N} L_{T_i,j} \times O_{T_i,j} \times a_{T_i,j} \\
\qquad\qquad a_{T_i,0} = 1 \\
a_{T_i,j} \in \{0,1\} \quad j = 1,2,\ldots,n \\
\qquad\qquad a_{T_i,b_j} \geq a_{T_i,j}
\end{cases}
\tag{39}
$$

where $P_{island}$ is the amount of power that can be supplied by the isolated island. $L_{T_i,j}$ is the load size. $O_{T_i,j}$ is the load weight. $a_{T_i,j}$ is a binary variable which means the state change parameter of the node $j$ during the period $T_i$. $a_{T_i,j} = 1$ indicates that the node is in the island and $a_{T_i,j} = 0$ indicates that the node is not in the island. $a_{T_i,0}$ is root node. $a_{T_i,b_j} \geq a_{T_i,j}$ is the node connection constraint, which means that if the child node is selected into the island, the parent node must also be selected into the island.

3. According to the initial island range, it is verified whether the island meet the real-time power balance through defense period. Therefore, in the third step, the power balance constraints are used to modify the initial island range to obtain a new island model which load requirements is determined and power balance is satisfied. In order to maintain the power balance in the island, the load power must be determined according to the DGs output prediction curve and charge power constraints, discharge power constraints, and capacity constraints of the ESS. In addition, the islands must also ensure power flow balance, as shown in Equation (38). Finally, the optimal island is established.

As shown in Figure 3, both the load power curve (the solid black line) and DGs curve (the black dotted line) can be predicted when $t \in (T_s, T_e)$. The ESS is charging when $t \in (T_s, T_1) \cup (T_2, T_e)$ and discharging when $t \in (T_1, T_2)$. The ESS can be considered as a load when it is in the charging state and as a power source when it is in the discharging state. Therefore, the following constraints must be subjected during the entire operation of the island, Table 1 is as follows:

**Table 1.** Island constraint.

| Constraints | Charging State | Discharging State |
|---|---|---|
| Island Power Balance | $P_{load} = \sum P_{DG}^t - P_{ch}^t$ | $P_{load} = P_{dis}^t + \sum P_{DG}^t$ |
| Charge and discharge power constraints | $P_{ch,min}^t < P_{ch}^t < P_{ch,max}^t$ | $P_{dis,min}^t < P_{dis}^t < P_{dis,max}^t$ |
| Island load demand | $E_{load} = \sum E_{DGs}$ | $E_{load} = \sum E_{DGs} + E_{ess}$ |
| State of charge (SOC) constraints | $E_{ess,min} < E_{ess} \le E_{ess,max}$ | $0 < E_{ess} \le E_{ess,critical}$ |

Where $P_{load}$ is load power, $\sum P_{DG}^t$ is actually used output power of the DGs, $P_{ch}^t$ and $P_{dis}^t$ are actual charge power and discharge power of the ESS, $P_{ch,max}^t$ and $P_{dis,max}^t$ are maximum charge power and discharge power of the ESS, $P_{ch,min}^t$ and $P_{dis,min}^t$ are minimum charge power and discharge power of the ESS, $E_{load}$ is island load demand. $\sum E_{DGs}$ and $E_{ess}$ are the amount of power supplied by DGs and ESS, $E_{ess,min}$ and $E_{ess,max}$ are the maximum capacity of the ESS, $E_{ess,critical}$ is the capacity of the ESS at the critical moment of charge and discharge. Therefore, it can be seen from Figure 3 that yellow dotted lines represent the sum of the maximum charge power of the ESS and load power. The red dotted line indicates the maximum power of the backup power supply (BPS) in island. The blue solid line is actually used DGs output. The purple line is actual discharging power of ESS.

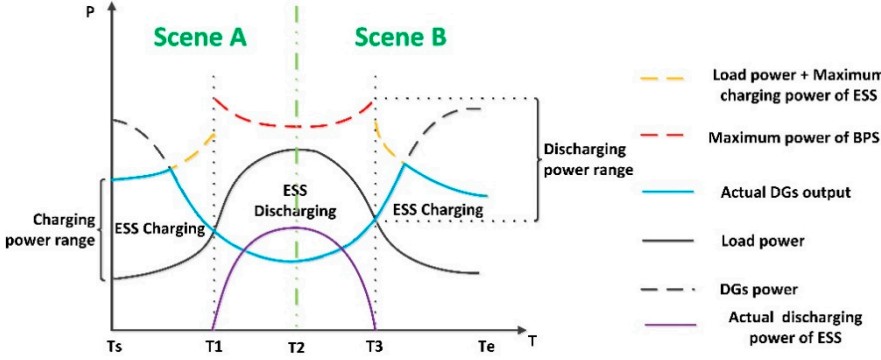

**Figure 3.** Power curve in island mode operation.

The DGs output and load power will fluctuate over time during the whole defense period which have two scenes appear alternately, thus the ESS energy transfer process of the following two scenes that subject to the constraints of Table 1 need to be analyzed. Please see Appendix B for specific details.

*4.3. CADS Evolution Process*

The optimization goal of CADS is:

$$min\Delta P_{k,cut} = P_{k,fur} - maxP_{k,tra} \tag{40}$$

In the formula, $P_{k,fur}$ is the load of the distribution line. $\Delta P_{k,cut}$ is the amount of loads shedding. $P_{k,tra}$ is the amount of protected load. Finally, the CADS is carried out according to the following steps, which are briefly described as follows:

(1) Search for high-risk distribution lines and determine the defense order according to the difference between the threshold and the risk value. At the same time, find all available interconnection switches and section switches for the distribution line;

(2) Transfer the load using an interconnection switch. This step attempts to use a interconnection switch to transfer all loads of the distribution line, if it can be achieved, go to step (6), otherwise go to step (3);

(3)  Transfer the load by using another interconnection switch. If only one interconnection switch cannot transfer all the loads of the distribution line, this step attempts to use other interconnection switches to transfer the remaining loads of the distribution line. If it can be realized, go to step (6), otherwise go to step (4);

(4)  Look for other defense strategies to make up for the shortcomings of a single solution. If the power flow direction of DG and the tie-line are the same, the DG can be used to assist when the loads cannot be completely transferred by the tie-lines, and if there are still no transfer loads, the island mode is used. If the island can be achieved, go to step (5);

(5)  If it can achieve complete defense, go to step (7), otherwise go to step (6);

(6)  If the ability of the active defense plan is exceeded after the above process, part of the loads must be cut off in order to ensure the safety constraints of the distribution network;

(7)  If there are other high-risk lines, go to step (1), otherwise, go to step (8);

(8)  Get an active defense evolution plan and release dispatch instructions to the next moment until the end of the disaster;

(9)  Determine if the ice disaster is over. If it is not over, go to step (1), otherwise the process end.

The evolution flow chart of the active defense strategy is shown in Figure 4.

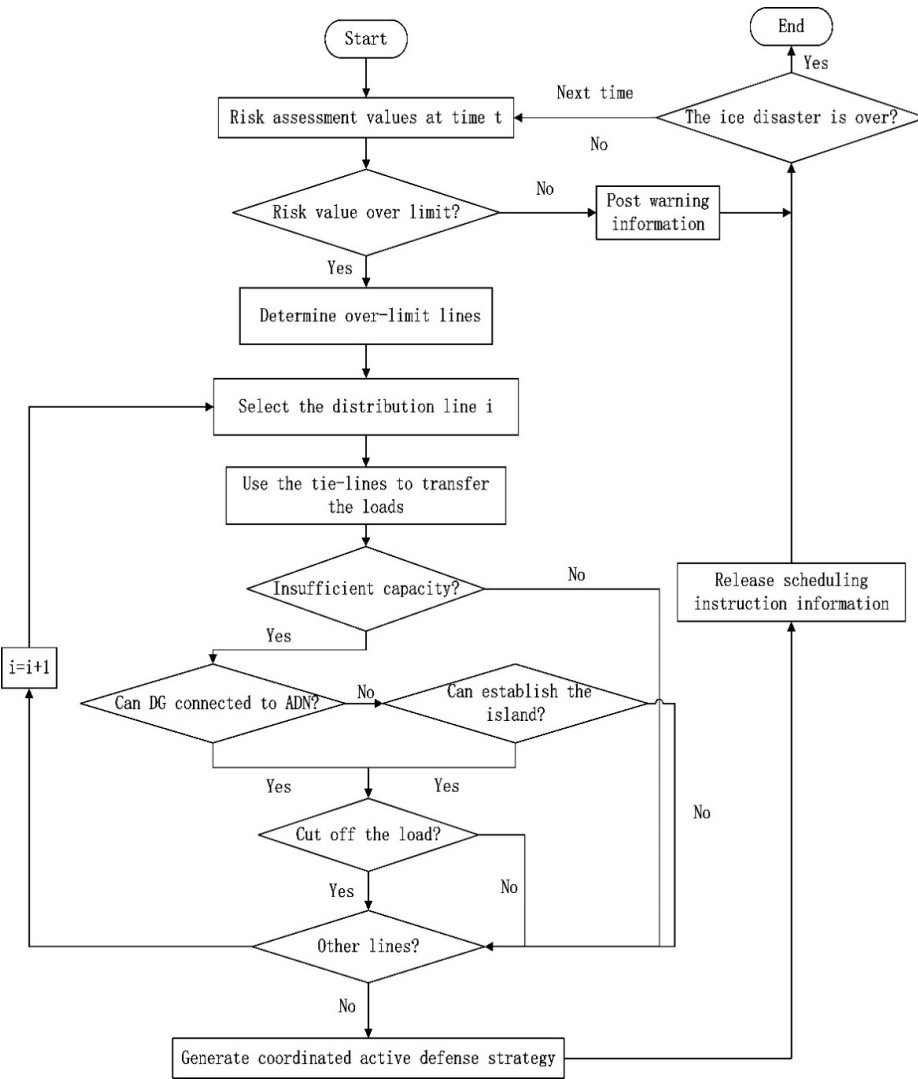

**Figure 4.** Evolution process of CADS.

According to the coordinated active defense evolution process, it can deal with the group faults risk and reduce the impact of accidents in the ADN.

## 5. Simulation Analysis

In this paper, the distribution network of a city in northeast China is used as a reference area. At 7:00 on 7 November 2018, freezing rain began to appear in the large city. The distribution lines began to trip and overlap at 11:30. After 15:00, the wind speed gradually expanded and the precipitation gradually decreases.

The distribution network of reference area consists of 4 substations (CG substation, Y substation, RB substation, and GZ substation), including a total of 74 nodes and 73 distribution lines with a voltage level of 66 kV, 4 BPS, and 4 tie-lines. Reference region power distribution system node diagram is shown in Figure 5. There are three types of nodes in the reference region power distribution system diagram. The yellow nodes are circuit breaker nodes, the blue nodes are branch nodes which can lead multiple feeders, and the black nodes are ordinary nodes which including load switches and 66/10 kV substation.

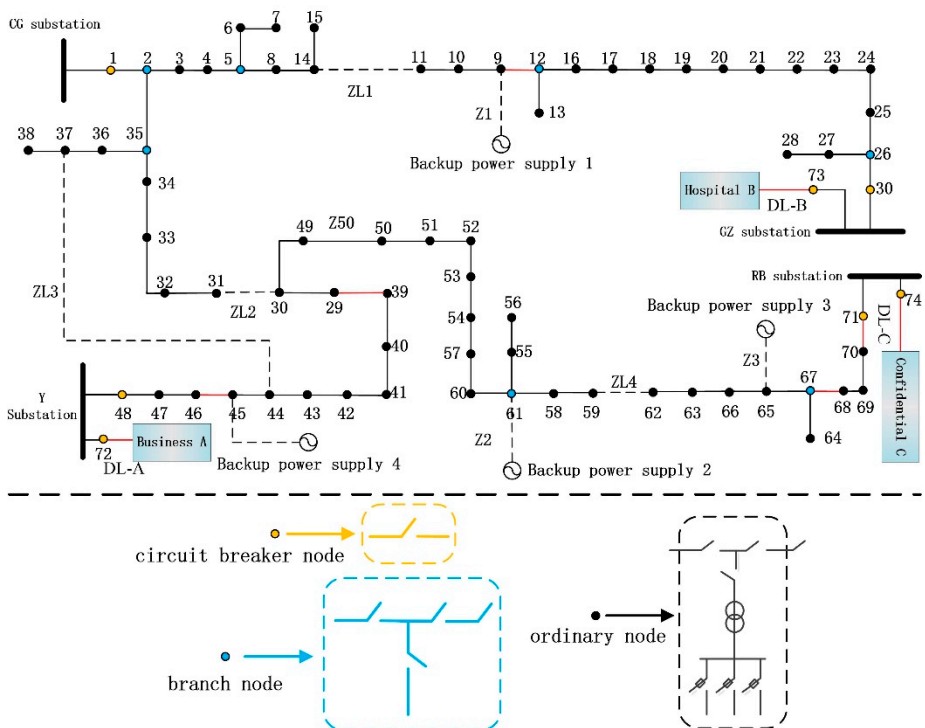

**Figure 5.** Reference area simplified wiring node diagram.

This paper selects 8 important distribution lines which are highlighted in red in the Figure 5. The basic information of the 8 lines are shown in Appendix C. Among them, three dedicated lines (DL) DL-A, DL-B, DL-C are used for business hospital and confidentiality, respectively. In addition, these BPSs are installed at nodes 9, 45, 61, and 65. ZL1–ZL4 are 4 tie line switches that used to transfer the loads. Z1–Z4 are switches of BPS which are opened in normally state. There is a segment switch between each two nodes such as Z50. In the normal state, ZL1–ZL4, Z1–Z4 are shown as dotted lines and segment switches are shown as solid lines. Among them, the solid line indicates the switch is closed and the dotted line means the switch is opened.

### 5.1. Distribution Network Vulnerability Analysis

Select historical data of ice disasters and historical fault data during 2008–2018, and meteorological data including the wind speed, rainfall, angle and other variables is simulated at 12:00. Finally, the

predicted values of comprehensive weak rate $F(k, t)$ of each distribution line in ice weather at 13:00 are shown in Table 2.

**Table 2.** Vulnerability analysis of distribution network in ice weather.

| Lines | $P_k^t(Y_2)$ | $P_k^t(Y_3)$ | $P_k^t(Y_4)$ | $P_k^t(Y_5)$ | $P_k^t(X)$ | $\eta_{k,1}^t$ | $\eta_{k,2}^t$ | $\eta_{k,3}^t$ | $\eta_{k,4}^t$ | $\eta_{k,5}^t$ | $\eta_{k,sit}^t$ | $F(k,t)$ |
|---|---|---|---|---|---|---|---|---|---|---|---|---|
| 45–46 | 0.125 | 0.04 | 0.027 | 0.009 | 0.239 | 0.25 | 0.459 | 0.214 | 0.425 | 0.3 | 0.238 | 0.420 |
| 67–68 | 0.150 | 0.045 | 0.027 | 0.004 | 0.275 | 0.625 | 0.307 | 0.214 | 0.54 | 0.525 | 0.343 | 0.518 |
| DL-B | 0.063 | 0.045 | 0.021 | 0.003 | 0.148 | 1 | 0.18 | 0.539 | 0.18 | 0.05 | 0.096 | 0.230 |
| 9–12 | 0.178 | 0.045 | 0.023 | 0.0096 | 0.265 | 0.325 | 0.395 | 0.388 | 0.74 | 0.5 | 0.364 | 0.534 |
| 70–71 | 0.240 | 0.039 | 0.029 | 0.0091 | 0.343 | 0.25 | 0.231 | 0.257 | 0.27 | 0.175 | 0.163 | 0.450 |
| 29–39 | 0.147 | 0.034 | 0.019 | 0.0037 | 0.223 | 0.375 | 0.325 | 0.366 | 0.45 | 0.47 | 0.311 | 0.465 |
| DL-A | 0.058 | 0.020 | 0.013 | 0.007 | 0.116 | 1 | 0.187 | 0.429 | 0.18 | 0.075 | 0.103 | 0.207 |
| DL-C | 0.071 | 0.039 | 0.015 | 0.005 | 0.138 | 1 | 0.151 | 0.217 | 0.18 | 0.075 | 0.076 | 0.204 |

The result shows:

(1) It can be seen from Table 2 that $F(9 - 12, t)$ and $F(67 - 68, t)$ are the highest. $\eta_{9-12,4}^t$, $\eta_{67-68,4}^t$, $\eta_{29-39,4}^t$, and $\eta_{45-46,4}^t$ are greatly affected by the ice weather. The distribution lines 45–46 and 9–12 contain the BPS which $\eta_{45-46,2}^t$ and $\eta_{9-12,2}^t$ are obviously high.

(2) As shown in Table 2, $\eta_{DL-B,3}^t$ and $\eta_{DL-A,3}^t$ are obviously insufficient. $\eta_{9-12,5}^t$, $\eta_{67-68,5}^t$, and $\eta_{29-39,5}^t$ are obviously high which indicates that the fault propagation of these three lines degree is large and easy to cause cascading faults. At the same time, distribution lines 67–68 and 29–39 occupy important positions in the topology, so $\eta_{67-68,1}^t$ and $\eta_{29-39,1}^t$ are also high.

(3) $P_k^t(X)$ and $\eta_{k,sit}^t$ are shown in Figure 6. Among them, the $P_{70-71}^t(X)$ is the highest, but the $F(70 - 71, t)$ is not the highest. It indicates that the failure rate of the distribution line in the ice disaster is not only affected by the external environment such as ice accretion, but also the internal operation situation of the distribution network is an important factor that affects the stable operation of the distribution line. Therefore, the comprehensive vulnerability rate is more reasonable.

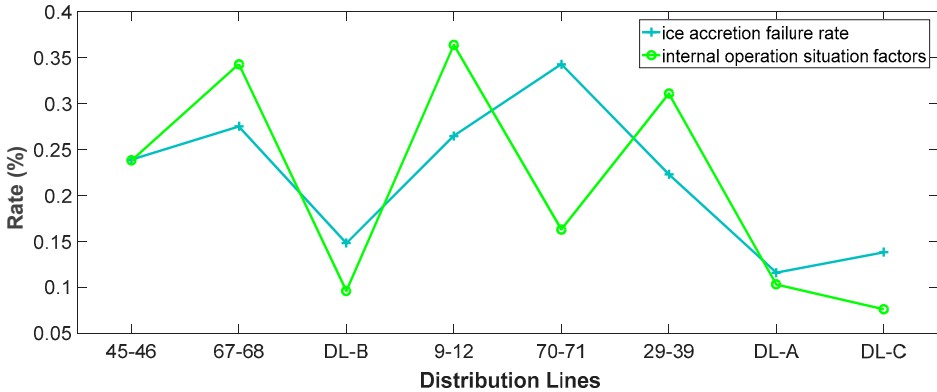

**Figure 6.** Two methods for predicting distribution line faults.

*5.2. Risk Assessment Simulation*

According to the meteorological data and power flow data at 12:00, combined with other relevant real-time data, a comprehensive risk assessment of the eight distribution lines at 13:00 is carried out. The risk assessment values are shown in Table 3 and the risk assessment value is $L = [0.450\ 0.623\ 0.329\ 0.719\ 0.516\ 0.522\ 0.360\ 0.260]$. The comprehensive vulnerability rate and risk values of each distribution line is shown in Figure 7. The three types of risk indicators for each distribution line are shown in Figure 8, as shown by data:

(1) It can be seen from Figure 7 that $F(k,t)$ is positively correlated with the $L_k$, and the difference in $L_k$ is more obvious than $F(k,t)$. The result of Figure 8 is calculated through a standardized process that shows the different impact of these three risk indicators on each distribution line. Therefore, different indicators can obtain a more comprehensive risk prediction result and achieve more accurate analysis of the operating conditions in the distribution network.

(2) It can be seen from Table 3 that $L_{DL-A}$, $L_{DL-B}$, and $L_{DL-C}$ are smaller than others, which are 0.360, 0.329, and 0.260 respectively. While $L_{9-12}$, $L_{67-68}$, and $L_{29-39}$ are larger than others which are 0.623, 0.719, and 0.522, respectively. It is necessary to adopt the defense strategy in time because it was verified that these three risk values exceeded the threshold.

(3) It can systematically perceive the status of distribution network in real-time by using the risk assessment method proposed in this paper, and gain valuable time for the staff to take the active defense strategy.

**Table 3.** Eight lines risk value results.

| Index | Weights | Line 45–46 | Line 67–68 | Line DL-B | Line 9–12 | Line 70–71 | Line 29–39 | Line DL-A | Line DL-C |
|---|---|---|---|---|---|---|---|---|---|
| $R(t_i)$ | 0.064 | 10.37 | 18.69 | 5.25 | 20.98 | 17.71 | 13.64 | 7.89 | 7.16 |
| $T_{fault}$ | 0.056 | 35.86 | 17.44 | 177.2 | 10.28 | 15.93 | 27.88 | 96.85 | 110.74 |
| $\phi$ | 0.048 | 0.479 | 0.64 | 0.364 | 0.705 | 0.656 | 0.493 | 0.383 | 0.378 |
| $r_a$ | 0.076 | 0.0204 | 0.0648 | 0.0216 | 0.077 | 0.015 | 0.038 | 0.027 | 0.025 |
| $r_b$ | 0.076 | 8.19 | 16.5 | 3.77 | 20.1 | 5.69 | 14.8 | 4.92 | 4.57 |
| $r_c$ | 0.076 | 3.25 | 10.6 | 4.49 | 3.75 | 7.83 | 9.26 | 3.88 | 5.03 |
| $\xi(k,t)$ | 0.096 | 0.025 | 0.052 | 0.046 | 0.043 | 0.027 | 0.047 | 0.041 | 0.041 |
| $C(k,t)$ | 0.096 | 0.091 | 0.169 | 0.252 | 0.136 | 0.094 | 0.159 | 0.246 | 0.245 |
| $e_1$ | 0.078 | 0.0146 | 0.027 | 0.024 | 0.027 | 0.0135 | 0.0405 | 0.0276 | 0.0675 |
| $e_2$ | 0.066 | 0.01 | 0.029 | 0.034 | 0.015 | 0.011 | 0.019 | 0.038 | 0.038 |
| $e_3$ | 0.066 | 0.35 | 0.22 | 0.41 | 0.27 | 0.5 | 0.18 | 0.38 | 0.45 |
| $e_4$ | 0.066 | 0.37 | 0.11 | 0.53 | 0.17 | 0.36 | 0.2 | 0.66 | 0.42 |
| $e_5$ | 0.068 | 0.009 | 0.018 | 0.018 | 0.009 | 0.009 | 0.018 | 0.018 | 0.018 |
| $e_6$ | 0.068 | 0.021 | 0.053 | 0.053 | 0.021 | 0.021 | 0.053 | 0.053 | 0.053 |

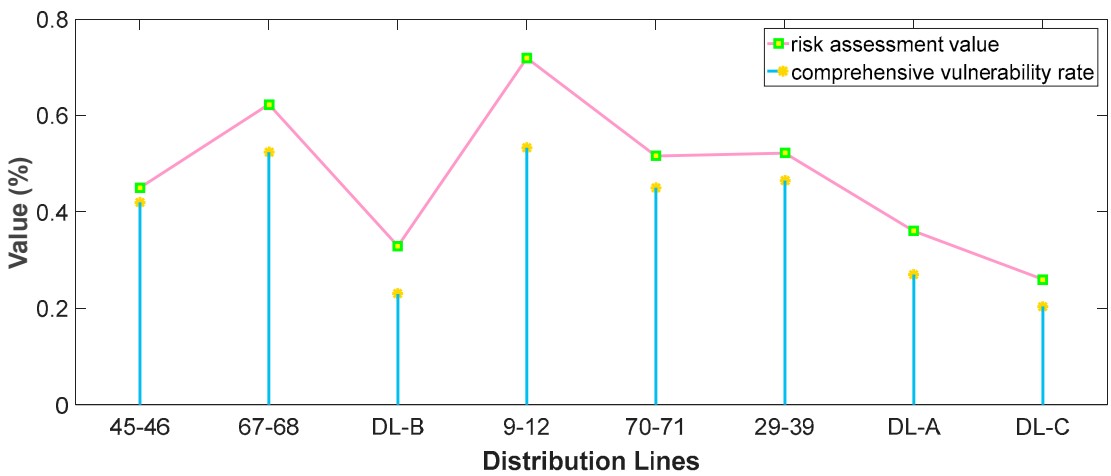

**Figure 7.** Comprehensive vulnerability rate and risk value comparison.

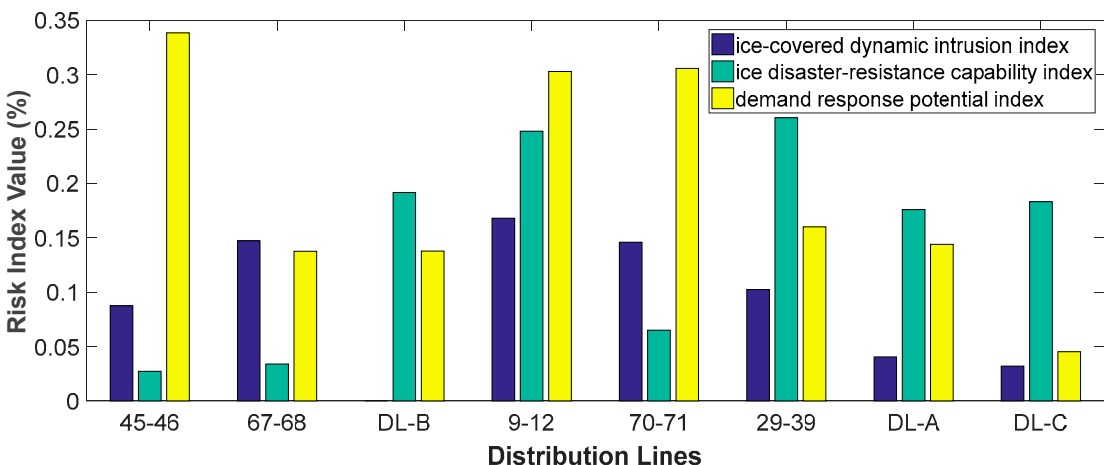

**Figure 8.** Three types of risk index of each distribution line after standardization.

## 5.3. Defense Strategy Simulation

According to the results of Sections 5.1 and 5.2, it is shown that the most likely fault locations are lines 9–12, 67–68, and 29–39. If no action is taken, the amount of loads loss will reach 294,320 kW, which will bring huge losses. In order to avoid the impact of the faults on the distribution network, the different strategies are adopted and simulated in this paper for comparison. Figures 9–11 are network structures of different defense strategies adopted before these lines fail. Among them, the red nodes represent loads blackout when the line fails, and the defense strategy cannot help them work normally. Yellow nodes indicate that if no strategy is taken, these nodes will also lose power, but the network topology can be changed through defense scheme to ensure the safe and stable operation of the loads. The blue line indicates that the tie line switch and BPS switch are closed.

The network structures with only the load transfer strategy and only the islands strategy are shown in Figures 9 and 10:

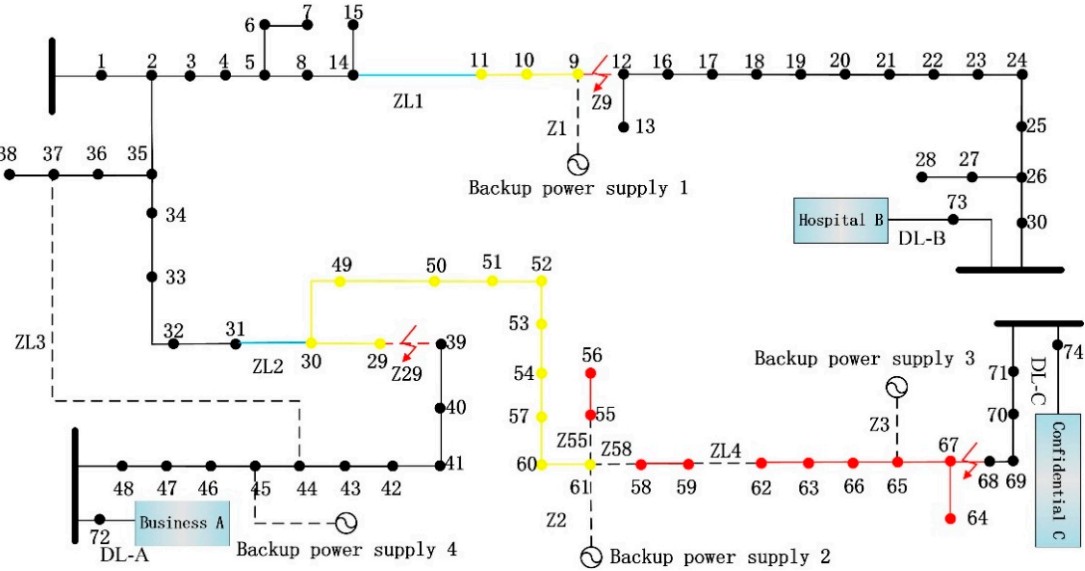

**Figure 9.** Network structure after transferring loads.

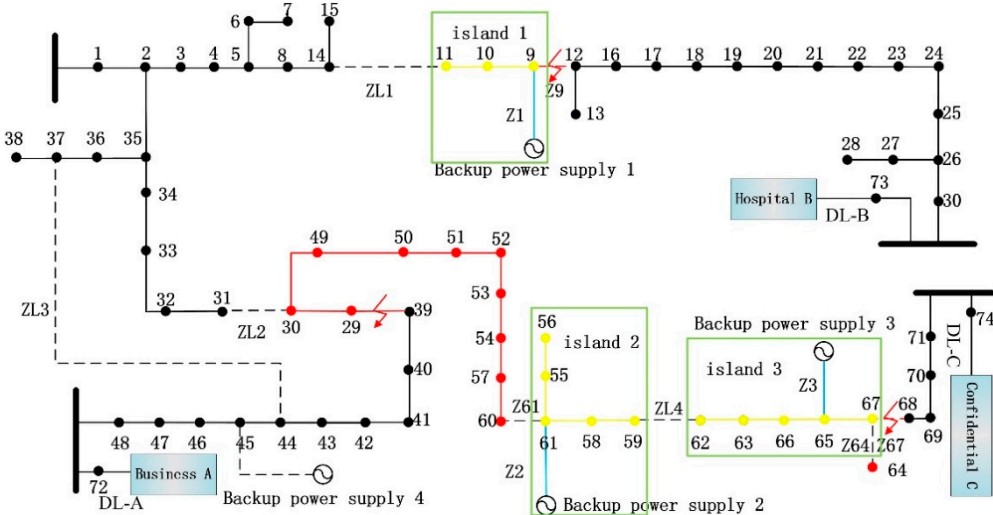

**Figure 10.** Network structure after islands strategy.

As shown in Figure 9, the tie line switches ZL1 and ZL2 are closed, and segment switches Z9, Z29, Z55, and Z58 are opened at the same time. Yellow nodes represent loads that can be transferred. The red nodes represent loads that have not been transferred, and all of these loads will lose power.

It can be seen from Figure 10 that Z1, Z2, and Z3 are closed, so island 1 consists of the BPS 1 and the loads of yellow nodes (9, 10, 11), island 2 consists of the BPS 2 and the loads of yellow nodes (55, 56, 58, 59, 61), island 3 consists of the BPS 3 and the loads of yellow nodes (62, 63, 65, 66, 67). The yellow nodes represent the loads that were incorporated into the island. Red nodes represent loads that have not been included in the island, which means all of these nodes will lose power once the line occurs faults.

The network structure which CADS adopted is shown in Figure 11:

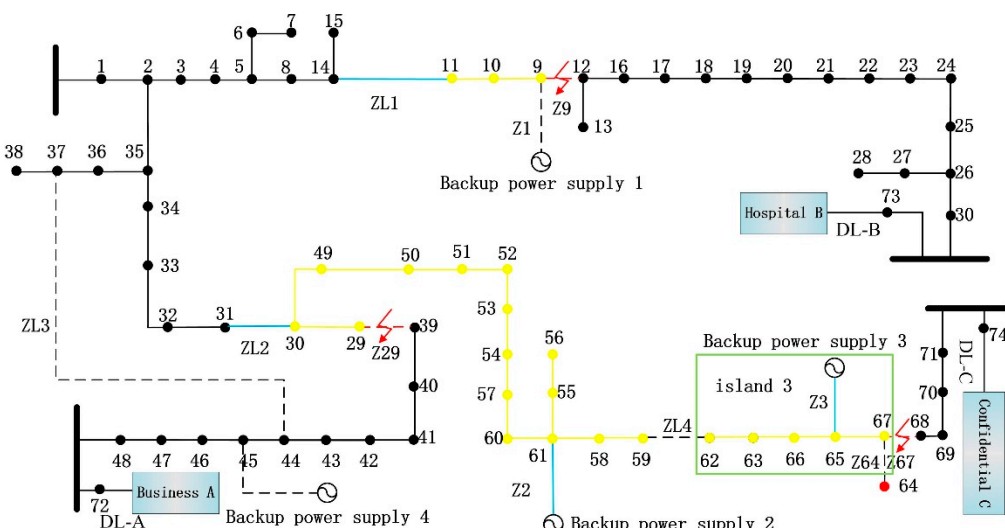

**Figure 11.** The network structure after CADS.

It can be seen from Figure 11 that after closing ZL1, ZL2, Z2, Z3 and opening Z9, Z29, Z64, Z67. When the CADS is adopted, the tie line ZL1 is used to transform nodes 9, 10, and 11, the island is established by the BPS 3 and nodes 62, 63, 65, 66, 67. At the same time, the tie line ZL2 and the BPS 2 cooperate with each other, the DG that connected the ADN and tie line ZL2 are used to defend the risk area.

Compare the results of the three defense strategies, as shown in Table 4.

Table 4. Comparison of three defense strategies.

| Defense Strategies | Switching Action Process | Number of Switching Operations | Loss of Load (10,000 kW) |
|---|---|---|---|
| no strategy | no | 0 | 29.432 |
| only the load transfer strategy | close ZL1, ZL2; disconnect Z9, Z29, Z55, Z58. | 6 | 16.514 |
| only the island strategy | access Z1, Z2, Z3; disconnect Z9, Z61, Z64, Z67. | 7 | 18.875 |
| coordinated active defense strategy (CADS) | access Z2, Z3; close ZL1, ZL2; disconnect Z9, Z29, Z64, Z67. | 8 | 1.61 |

As shown in Table 4, the number of switch operations is the least when the load transfer strategy is adopted. However, the capacity of the tie-line ZL2 is limited, the loads are transferred to node 61 at most, so the load loss is much larger than the CADS. On the contrary, all nodes can avoid the danger of power failure except 64 if the CADS is used. When the island strategy is adopted, the number of switches operations is more than the load transfer strategy. Furthermore, because of the limited capacity of the BPS 2, there are so many red nodes that cannot be included in the islands in Figure 10. Similarly, the node 64 cannot be protected by the BPS 3, it makes it that the load loss is largest among three strategies.

Through simulation analysis, the advantage of using this strategy is that load loss is minimized. Moreover, the maintenance department can use the ice melting device to melt ice on high-risk lines or use artificial de-icing teams to reduce the risk of these lines. Since the goal of active defense is to pursue the least amount of load loss under certain constraints, the method proposed in this paper is more reasonable in the natural disaster environment.

## 6. Conclusions

In order to improve the safety and stability of ADN under ice weather, a series of research is carried out in this paper from three aspects: fault rates prediction, risk assessment, and active defense. The comprehensive vulnerability rate model is established and the risk assessment system is proposed which improve the ability of monitoring and early warning in the ice weather. Furthermore, the coordinated active defense strategy (CADS) considering risk pre-judging is designed by using the distributed energy and tie-lines. The application of this strategy not only minimizes load loss, but also provides scientific decision-making and reduces the impact of fault in ice weather.

**Author Contributions:** X.-R.L. conceived the idea for the manuscript and wrote the manuscript with input from H.W., Q.-Y.S., and P.J. All authors have read and agreed to the published version of the manuscript.

**Funding:** This work is supported by National Key R&D Program of China under grant (SQ2018YFA0702200).

**Conflicts of Interest:** The authors declare no conflict of interest.

## Appendix A

Assume that the probability of each uncertain event is independent of each other, the full probability formula can be used to calculate the $P_k^t(Y)$. {Y1, Y2, Y3, Y4, Y5} is defined as a division of the event space set $\Omega$, and define "faults directly related to ice icing" as event $X$, as shown in Figure A1.

The distribution line disconnection event and the inverted tower event are inseparable from the stress conditions of the overhead line and the tower. Therefore, it is necessary to perform mechanical analysis on the wire and the corresponding tower. In order to simplify the calculation workload, the distribution line can be regarded as the ideal cable. Therefore, the internal stress $\sigma_0$ can be obtained according to the parabolic formula. The specific calculation process is shown in Equations (A1)–(A7):

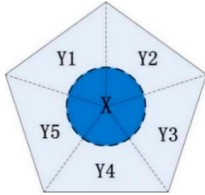

**Figure A1.** Schematic diagram of $\Omega$ division of ice fault events space.

$F_1$ and $F_2$ are the wind load changes caused by the horizontal and vertical wind. The corresponding calculation formulas are shown as:

$$F_1 = 10^{-3}\frac{\rho_w}{2}V_1^2\left(2R_d^j + D\right)L_t \tag{A1}$$

$$F_2 = 10^{-3}\frac{\rho_w}{2}V_2^2\left(2R_d^{j-1} + D\right)L_t \tag{A2}$$

In the formula, $\rho_w$ is the air density value, the unit is $g/cm^3$ , which can be obtained from the measured atmospheric pressure. $V_1$ and $V_2$ represent the horizontal and vertical components of the wind speed, the unit is m/s. $R_d^j$ is the thickness of the ice accretion. $D$ is the radius of the distribution line.

The self-weight load $G$ of the distribution line is:

$$G = \frac{10^{-3}}{4}\pi D^2 L_t \rho_1 g_n \tag{A3}$$

where $\rho_1$ is the density of the line itself, the unit is $g/cm^3$.

Thus, the predicted value of total load $H$ can be calculated by the Equation (28), and the total specific load $\gamma$ of the distribution line is:

$$\gamma = \frac{H}{\frac{1}{4}\pi D^2 L} \tag{A4}$$

In the formula, $L$ is the distribution line length, the unit is *m*.

The line stress $\sigma_n$ can be calculated by using the force state equations, which are shown as:

$$\begin{cases} \sigma_n^2(\sigma_n + A) = B \\ A = \frac{Kg_n^2 l^2}{24\sigma_m} - \sigma_m + \alpha E(\theta_n - \theta_m) \\ B = \frac{E\gamma^2 L_t^2}{24} \end{cases} \tag{A5}$$

In the formula, set the subscripts *m* and *n* are the initial state and current state of the line, respectively. Among them, $\theta_n$ and $\theta_m$ respectively represent the current temperature of the line and the initial temperature during normal operation. $\sigma_n$ and $\sigma_m$ respectively represent the stress that the line is subjected at the current temperature and during normal operation. In addition, *K* is the coefficient of elasticity of the line, the unit is $N/mm^2$. $\alpha$ is the coefficient of line temperature expansion. Therefore, the internal stress $\sigma_0$ can be calculated by using $\sigma_n$ from the Equation (A5) in the manuscript.

If the heights of the two towers are different, higher towers are subject to greater stress at the distribution line interface, and the high difference angle $\varphi$ should be considered. The projection distance $l_m$ between the distribution line at the lowest point of the line and the top of the higher tower on the horizontal plane is:

$$l_m = \frac{l}{2}\frac{\sigma_n}{\gamma}\sin\varphi \tag{A6}$$

The internal stress $\sigma_0$ of the distribution line at the connection of the higher tower is:

$$\sigma_0 = \sqrt{\sigma_n + \frac{l}{4}\frac{\sigma_n^2}{\gamma^2}tan^2\varphi} \tag{A7}$$

After the maximum stress of the distribution line is obtained, the function is used to compare the maximum stress $\sigma_0$ with the distribution line stress upper limit $\sigma_s$, and the probability of the distribution line disconnection event is obtained:

$$P_{kj}^t(Y_2) = \begin{cases} 0 & \sigma_0 < 0 \\ K_{ice}e^{\frac{\sigma_0}{T_{ice}}} & 0 \le \sigma_0 < \sigma_s \\ 1 & \sigma_s \le \sigma_0 \end{cases} \tag{A8}$$

$$T_{ice} = \frac{\sigma_s}{ln\left(\frac{1}{K_{ice}}\right)} \tag{A9}$$

In the formula, $\mu_1$ is the safety factor, which can be taken as $\mu_1 = 2.5$. $K_{ice}$ is a constant, which is selected according to statistical data and distribution line fragility.

When the line is covered with ice, the tower may collapse due to ice accretion imbalance stress $\Delta F$ on the distribution lines exceeding the design upper limit:

$$\Delta F = \frac{\pi n D^2}{4}\sqrt{\sigma_1^2 + \sigma_2^2 - 2\sigma_1\sigma_2cos\theta_z} \tag{A10}$$

The probability of the inverted tower event is:

$$P_{kj}^t(Y_3) = \begin{cases} 0 & \sigma_0 < 0 \\ K_2e^{\frac{\Delta F}{\mu_2 F_T}} & 0 \le \sigma_0 < \sigma_s \\ 1 & \sigma_s \le \sigma_0 \end{cases} \tag{A11}$$

For distribution line dancing events, the vulnerability of the distribution line and the degree of asymmetry of the section increase with the ice accretion, so the "thickness coefficient of ice accretion" $\omega$ is proposed. The formula is as follows:

$$\omega = \frac{R}{40} \tag{A12}$$

In summary, the range of line dancing can be simplified to a function only related to ice thickness $R$, wind speed $V$, and angle $\beta$:

$$z = \widetilde{A}\sqrt{\left(1 - \widetilde{B}\omega_d\omega_r R\right)}(V - 4)sin(\theta - 45°) \tag{A13}$$

where $\widetilde{A}$, $\widetilde{B}$ are the coefficients related to distribution line length, radius, cross-sectional area, etc.

The probability of the distribution line dancing events is:

$$P_{kj}^t(Y_4) = \begin{cases} 0 & z < z_1 \\ \frac{z - z_1}{z_2 - z_1} & z_1 \le z < z_2 \\ 1 & z \ge z_2 \end{cases} \tag{A14}$$

where $z_1$, $z_2$ mainly depend on the cross-sectional area, the tension level of the length distribution line.

The relationship between the insulator flashover voltage $U$ and the density of the insulator salt before ice accretion $\rho_{SDD}$, altitude $H$, and ice weight $M$ is:

$$U = C\rho_{SDD}^{-e}\sigma_{20^\circ C}^{-d}(1 - \frac{H}{45.1})^{3.216}(1 - \frac{M}{45.1})^{4.36} \tag{A15}$$

where $C$ is the coefficient related to the insulator type and the quantity. $e$ is the pollution characteristic index. $\sigma_{20\,^\circ C}$ is the conductivity of the water at 20 °C. $d$ is a characteristic index affected by the conductivity of water. During the icing weather, $\rho_{SDD}$ and $H$ are unchanged, then $U$ is only related to the insulator ice weight $M$:

$$U = \overline{A}(1 - \overline{B}W)^{4.36} \tag{A16}$$

where $\overline{A}, \overline{B}$ are the coefficients related to the salt density, elevation, insulator type, insulator quantity, and density of ice.

The probability of insulator flashover event is:

$$P_{kj}^t(Y_5) = \begin{cases} 0.01 & U \le U_0 \\ K\,exp(\frac{U_0 - U}{T}) & U > U_0 \end{cases} \tag{A17}$$

where $U_0$ is the reference voltage and $K,\ T$ are coefficients.

## Appendix B

Scene A: When $t \in (T_s, T_1)$, the output of DGs (black dotted line) is greater than the load power (black solid line), and ESS is in the charging process. When $t \in (T_1, T_2)$, the output of DGs is less than the load power demand and ESS is in the discharging process.

Scene B: When $t \in (T_2, T_3)$, the output of DGs (black dotted line) is less than the load power (black solid line) and ESS is in the discharging process. When $t \in (T_3, T_e)$, the output of DGs is greater than the load power and ESS is in the charging process.

Since the ESS is subject to charge and discharge power constraints and capacity constraints at all times, we try to divide scene A into three different cases to analyze the energy transfer process of ESS.

Case A1: As shown in Figure A2, assuming that the ESS is not fully charged at the start time $T_s$ of scene A, the ESS is charging with the maximum charging power until time $t_{m1}$ and fully charged at that time. When $t \in (t_{m1},\ T_1)$, actual DGs output power curve is indicated by the blue line. There is excess output of DGs that cannot be absorbed by the ESS during the whole charging process, thus a part of the energy is wasted, which is indicated by a shaded part in Figure A2. When $t \in (T_1, T_2)$, DGs and ESS jointly supply power to the load. The red dotted line indicates the maximum power of the backup power supply (BPS) in island. The purple line is actual discharging power of ESS.

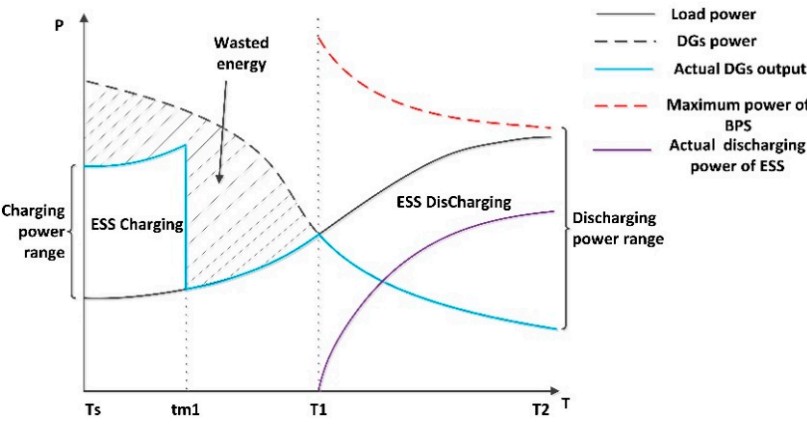

**Figure A2.** Case A1.

Case A2: As shown in Figure A3, assuming that the ESS is not fully charged at the start time $T_s$ of scene A, the ESS is charged with the maximum charging power before $t_{m2}$. However, because of the output limitations of DGs, ESS failed to charge with maximum power until fully charged at time $t_{m3}$. Similarly, the energy which wasted is indicated by a shaded part in Figure A3. When $t \in (T_1, T_2)$, DGs and ESS jointly supply power to the load.

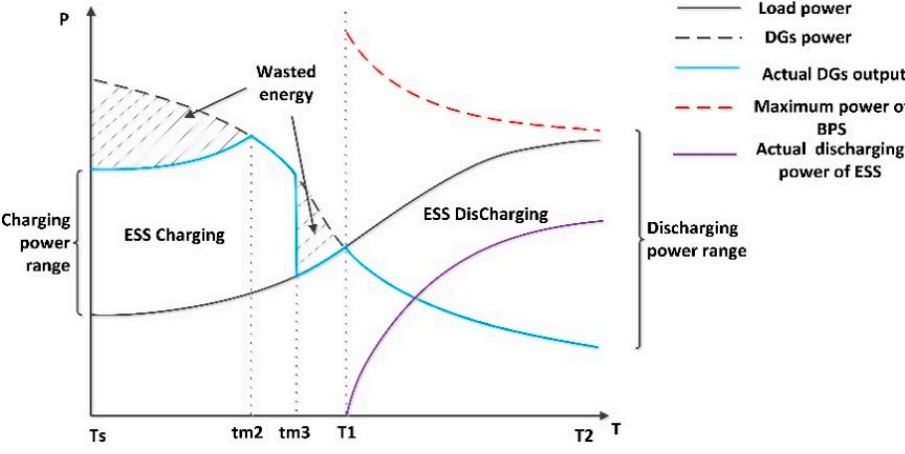

**Figure A3.** Case A2.

Case A3: As shown in Figure A4, assuming that the ESS is not fully charged at the start time $T_s$ of scene A. Due to the output limitations of DGs, ESS failed to charge with maximum power during the whole charging process until fully charged at time $t_{m4}$. Similarly, the energy which wasted is indicated by a shaded part in Figure A4. When $t \in (T_1, T_2)$, DGs and ESS jointly supply power to the load.

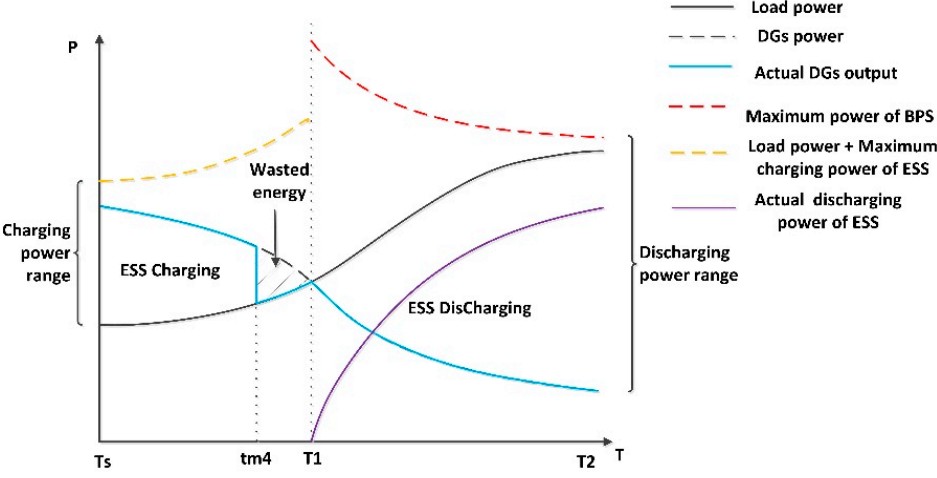

**Figure A4.** Case A3.

We also try to analyze the energy transfer process of scene B through the three cases.

Case B1: As shown in Figure A5, when $t \in (T_2, T_3)$, DGs and ESS jointly supply power to the load. Between $T_3$ and $t_{m5}$, the ESS is in a charging state but cannot be charged at maximum power because of the limitation of the DGs output until it is fully charged at time $t_{m5}$. Similarly, the energy which wasted is indicated by a shaded part in Figure A5.

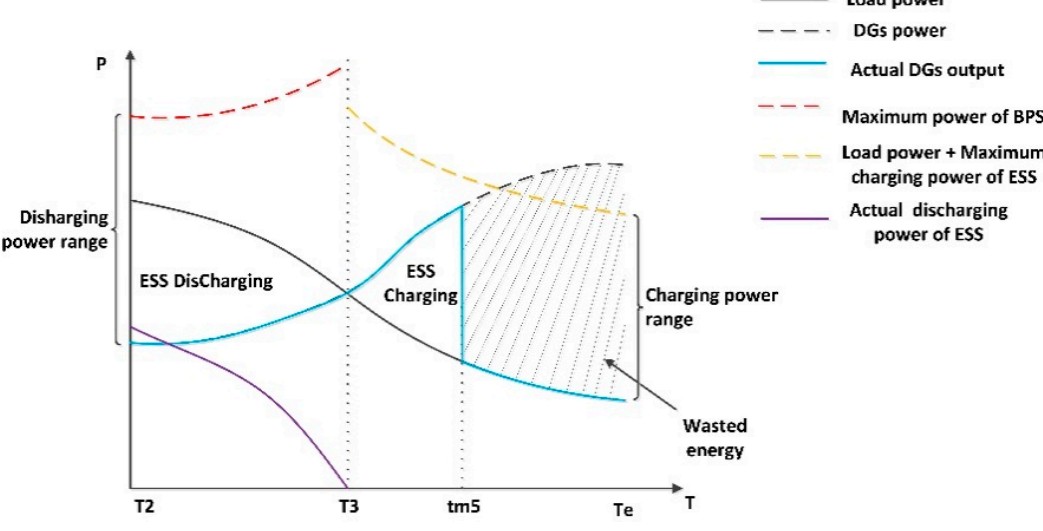

**Figure A5.** Case B1.

Case B2: As shown in Figure A6, when $t \in (T_2, T_3)$, DGs and ESS jointly supply power to the load. Between $T_3$ and $t_{m6}$, the ESS is in a charging state but cannot be charged at maximum power because of the limitation of the DGs output. Between $t_{m6}$ and $t_{m7}$, due to the ESS maximum charging power constraint, it can only be charged at the maximum power until eventually fully charged at $t_{m7}$. The excess power generated by the DGs is wasted when $t \in (t_{m7}, Te)$.

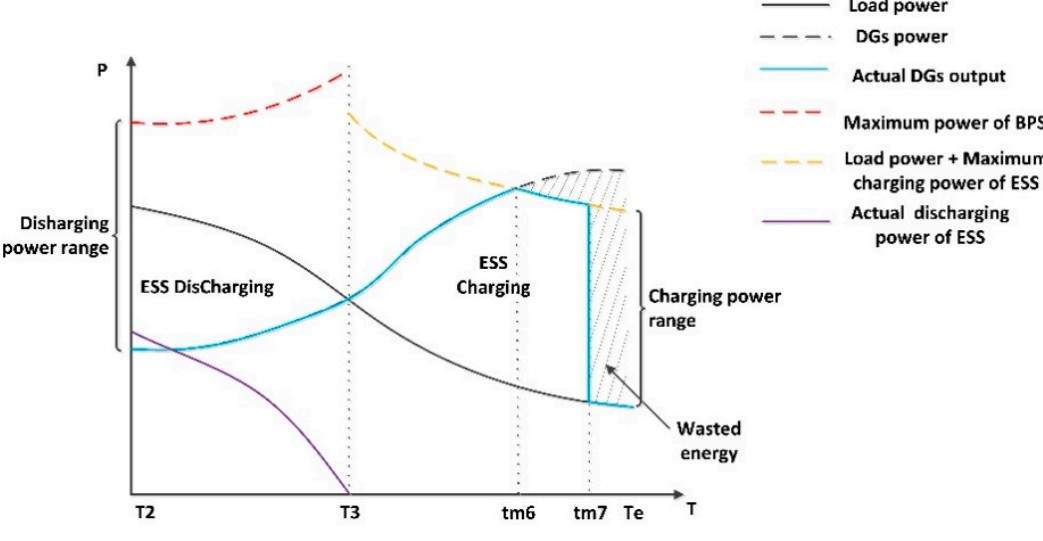

**Figure A6.** Case B2.

Case B3: As shown in Figure A7, when $t \in (T_2, T_3)$, DGs and ESS jointly supply power to the load. Between $T_3$ and $t_{m6}$, the ESS is in a charging state but cannot be charged at maximum power because of the limitation of the DGs output. Between $t_{m6}$ and $T_e$, due to the ESS maximum charging power constraint, it can only be charged at the maximum power. At the end time $Te$ of scene B, the ESS was still not fully charged. The energy which wasted is indicated by a shaded part in Figure A7.

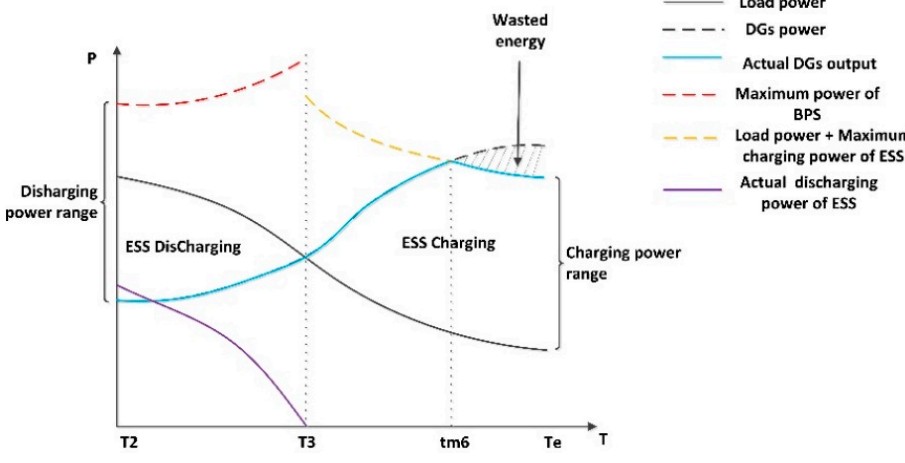

**Figure A7.** Case B3.

**Appendix C**

**Table A1.** Basic information of typical important user lines.

| Lines | Important User Corresponding Node | Length (m) | Line Type | Wire Cross-Sectional Area (mm²) |
|-------|-----------------------------------|------------|-----------|----------------------------------|
| 45–46 | 45 (confidential A) | 481 | JKLYJ-120 | 120 |
| 67–68 | 67 (confidential B) | 506 | JKLYJ-240 | 240 |
| DL-B | 73 (hospital B) | | Dedicated line | |
| 9–12 | 9 (hospital A) | 278 | JKLYJ-240 | 240 |
| 70–71 | 70 (confidential D) | 215 | JKLYJ-120 | 120 |
| 29–39 | 29 (financial A) | 450 | JKLYJ-150 | 150 |
| DL-A | 72 (business A) | | Dedicated line | |
| DL-C | 74 (confidential C) | | Dedicated line | |

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
