# Peer review of "Research on Dynamic Risk Assessment and Active Defense Strategy of Active Distribution Network under Ice Weather"

_applsci, doi:10.3390/app10020672_

Round 1

Reviewer 1 Report

The ADN, CADS, DG, ESS, should be defined at the first appearance and only one time. CADS is defined twice. 

Although the calculations are just statistical analysis, the geografic data should be explained concerning Fig. 4 and 8, 9, 10. Otherwise this paper is quite difficult to understand. Please explain the modelled area.

The island mode needs the detailed balance of the power plant and the consumer, energy storage system inside the isolated grid. This part can be discussed in a separate paper.  

English expressions like in page 2, line 6, "some relevant ideas and technical analysis of security defence were proposed in 17 - 20, "  ends with comma.

the next sentence should be connected with "and" or "in which", etc. to  "one is the strategy based on DC ice melting technology.   Such queer expressions can be found through the manuscript. For example, in page 3 line 19, Finally, define "faults" should be "defining" ot "the definition of".    

Reviewer 2 Report

Paper has merits but some improvements such as follows are recommended to be considered:

- please rephrase: "Therefore, in order to improve the adaptability of active distribution network in ice weather, the main contributions introduced by this paper from three aspects:"

- on page 2; also, on the same page (2), "The multisource situational awareness method is used to analyzes the internal operation situation of the distribution network in this paper, define the network topology situation, the power supply situation, the defense resource, the power flow situation and cascading fault factor, thus the comprehensive vulnerability rate model is established." needs a revision ("analyzes");

- on page 2, please revise "Besides, define the threshold of risk assessment to provide a complete data support and theoretical basis for the distribution network to enter the active defense cycle." the meaning is not clear. please define all the terms from equations 1,2,3 (R, ρ0);

- please rephrase " The probability that the icing event can be formulated as:" on page 3;  

- there are two paragraphs noted 2.3.1, one on page 3, another on page 4;

- "? is the radiation intensity of the surface of the battery.", is this term relevant if it is correct?

- in formula 13 it is not clear when the 3 cases appear;

- use upper case as first letter of paragraph 2.4 title;

- please review the numbering of the equation in the entire paper;

- on page 18, "Therefore, the internal stress can be obtained according to the parabolic formula" please provide the formula or refer to an equation;

- on page 18, it is not clear if the following statement is correlated with formula (24) or with another one; some terms are missing and others do not appear in the mentioned formula: "In the formula, ? is the ice load on the distribution line. ?1 and ?2 are the wind load changes caused by the horizontal and vertical wind. ? is the line length, the unit is m. ? is the self-heavy load of the distribution line."

Round 2

Reviewer 1 Report

My suggestion is,

Explaing the geographycal background of the  block diagram of the grid. Some expressions in the conditional mode is not connected to the main sentence. Isolated island model is better discccussed separately, because the detailed balance of the source and load, sometimes energy storage system is required.  

Reviewer 2 Report

Paper can be published in its actual form. 

Author Response

We appreciate your support and recognition.

Round 3

Reviewer 1 Report

I think the problems are improved so so.